# Attention as Inference via Fenchel Duality

## Abstract

Attention has been widely adopted in many state-of-the-art deep learning models. While the significant performance improvements it brings have attracted great interest, attention is still poorly understood theoretically. This paper presents a new perspective to understand attention by showing that it can be seen as a solver of a family of estimation problems. In particular, we describe a convex optimization problem that arises in a family of estimation tasks commonly appearing in the design of deep learning models. Rather than directly solving the convex optimization problem, we solve its Fenchel dual and derive a closed-form approximation of the optimal solution. Remarkably, the solution gives a generalized attention structure, and its special case is equivalent to the popular dot-product attention adopted in transformer networks. We show that T5 transformer has implicitly adopted the general form of the solution by demonstrating that this expression unifies the word mask and the positional encoding functions. Finally, we discuss how the proposed attention structures can be integrated in practical models.

## 1 Introduction

Attention-based deep neural networks are now integrated into cutting-edge language models that have revolutionized a broad range of tasks: machine translation [1, 15], sentiment classification [27], image captioning [29] and unsupervised representation learning [5], etc. Especially, attention plays a pivotal role in the construction of the transformer architecture [25], which has had a profound impact on the deep learning field.

Despite great empirical success, the driving principles of attention are still poorly understood. This lack of understanding impedes practitioners from confidently and appropriately using attention layers and makes it challenging to develop new attention-based neural architectures.

In this paper, we offer a new perspective for understanding attention by showing that it is in fact a solver for a certain type of optimization problem that corresponds to an inference task. We give several examples, all of which can be characterized as follows: given 1) an unreliable estimate of the mean of an unknown distribution $p$ on $\mathbb{R}^d$ and 2) a preference distribution $u$ on $\mathbb{R}^d$ encoding beliefs on $p$'s selection, the inference task is to get a better estimate of $p$'s mean given its unreliable estimate and $u$. We derive a convex optimization problem that is abstracted from the task and solve it by instead solving its Fenchel dual [22, p.104]. Remarkably, the derived expression of the improved estimate of $p$ gives a generalized attention structure whose special case is equivalent to the popular dot-product attention [15] that is also applied in the transformer network [25]. In addition, we show that our generalized attention expression has been implicitly adopted by T5 transformer [19] as the expression unifies the concept of word masks and its positional encoding functions. Extra examples are given to show how the generalized attention structures can be used in practice.

Submitted to 35th Conference on Neural Information Processing Systems (NeurIPS 2021). Do not distribute.

## 2 Related Works

Since 2019, several authors have investigated the properties and working mechanism of attention. This series of works mainly addresses whether the attention mechanism can serve as a proxy of saliency [9, 18, 23, 24, 26, 28]. Most of these works obtain insights into the attention mechanism by performing empirical studies. The related methods include analyzing the behaviours of trained attention-based networks [4], or pruning a few heads, or analyzing the effects of altering the attention weights [18, 26], or a mixture of these [9, 24].

Apart from understanding attention empirically, some theoretical results presented by Brunner el al. [3] and Hahn [7] show that the self-attention layers are not identifiable. This implies there could exist multiple combinations of attention weights that can provide equally good final predictions. In particular, such non-uniqueness means that the use of attention may complicate interpretability. Another important approach to understand attention is to analyze its asymptotic behaviour when the number of heads and the network width approach infinity [8, 30]. In this limiting case, the entire network can be seen as a Gaussian process [13] and its behaviours can be characterized by closed-form expressions that are not available in the finite case.

Very recently (since 2021) several theoretical works have appeared that study attention outside the asymptotic regime. Lu et al. [14] set up a simple attention-based classification model and derive a closed-form relationship between the word's embedding norm and the product of its key and the query. They empirically show that such relationship also exists in a more complicated and practical configuration. Ramsauer et al. [20] construct an equivalence relationship between attention and a newly proposed Hopfield network with continuous states. In particular, they show that the new Hopfield network's update rule is equivalent to the attention mechanism used in transformers [25].

## 3 A Motivating Example

We first consider a seemingly unrelated example, to illustrate the key ingredients of this paper.

Assume a probability distribution $p$ on $\mathbb{R}^d$ has a spherical Gaussian prior $u \sim \mathcal{N}(\mu, I_d)$. Let $h_p$ denote the mean of the unknown $p$. Given an unreliable observation $b$ of $h_p$, what is the best guess of $h_p$? To solve this problem, we may formulate the following optimization problem

$$p^* = \arg\min_p \frac{\alpha}{2} \left\| b - \int \mathbf{a} p(\mathbf{a}) \, d\mathbf{a} \right\|^2 + \mathcal{K}(p, u), \tag{1}$$

with $\alpha > 0$ responsible for the relative strength of the two terms, where $\mathcal{K}(p, u)$ denotes the KL divergence between $p$ and $u$. The basic idea behind (1) is that: although $b$ is not reliable, it should not be too far from $h_p = \int \mathbf{a} p(\mathbf{a}) \, d\mathbf{a}$. Also, as $u$ encodes the preferred value of $p$, we add the KL divergence term to show preference for $p$ that is close to $u$. As will be discussed later, such a formulation can be either obtained from the maximum likelihood principle or from the maximum entropy principle [10, 11]. In particular, Rioux et al [21] develop (1) for image de-blurring by applying Maximum Entropy on the Mean (MEM), an information-theoretic method due to Gamboa [6] but not yet widely known in machine learning.

After obtaining the minimizer $p^*$ of (1), its mean $\int \mathbf{a} p^*(\mathbf{a}) \, d\mathbf{a}$ gives our estimate of $h_p$. Rioux et al. [21] prove, via Fenchel duality [22, p.104] that the minimizer $p^*$ takes the form

$$p^*(\mathbf{a}) = \frac{u(\mathbf{a}) \exp\langle \mathbf{a}, \lambda^* \rangle}{\int u(\mathbf{a}') \exp\langle \mathbf{a}', \lambda^* \rangle \, d\mathbf{a}'}, \tag{2}$$

where

$$\lambda^* = \arg\max_{\lambda \in \mathbb{R}^d} \langle b, \lambda \rangle - \frac{1}{2\alpha} \|\lambda\|^2 - \log \int_{\mathbb{R}^d} u(\mathbf{a}) \exp\langle \mathbf{a}, \lambda \rangle \, d\mathbf{a}. \tag{3}$$

Note that $\int_{\mathbb{R}^d} u(\mathbf{a}) \exp\langle \mathbf{a}, \lambda \rangle \, d\mathbf{a} = \exp(\langle \mu, \lambda \rangle + \frac{1}{2} \|\lambda\|^2)$ as it is the moment generating function (MGF) of $u \sim \mathcal{N}(\mu, I_d)$. Substituting the expression into (3) followed by setting the derivative with respect to $\lambda$ to zero yields $\lambda^* = \frac{\alpha}{\alpha+1}(b - \mu)$. By (2), $p^*(\mathbf{a}) \propto \exp(-\frac{1}{2} \|\mathbf{a} - \mu\|^2 + \langle \mathbf{a}, \lambda^* \rangle) \propto \exp(-\frac{1}{2} \|\mathbf{a} - (\mu + \lambda^*)\|^2)$. Substituting $\lambda^* = \frac{\alpha}{\alpha+1}(b - \mu)$ into it implies that $p^*$ follows a Gaussian distribution $\mathcal{N}(\frac{1}{1+\alpha}\mu + \frac{\alpha}{1+\alpha}b, I_d)$. Thus, our estimate of $h_p$ is $\frac{1}{1+\alpha}\mu + \frac{\alpha}{1+\alpha}b$.

In this paper, we focus on a similar optimization problem that estimates $h_p$ assuming that $u$ is instead a discrete distribution. We show that such optimization problems naturally and frequently arise in neural network designs. By solving the optimization problem, we derive a closed-form approximation for the estimate of $h_p$, via Fenchel duality. The approximation then gives a generalized attention layer structure as shown in Fig 1. A special case of it is equivalent to the familiar dot-product attention [15] that is also adopted in transformers [25]. Moreover, we will show that T5 transformer [19] implicitly adopts our generalized attention expression.

# 4 Setup of a Design Problem

Throughout the rest of the paper, we consider a machine learning problem in which the objective is to predict an output quantity $Y$ from a given input $X$. Additionally, $Y$ may include $K$ components, namely, be expressed as $(Y^{(1)}, Y^{(2)}, \ldots, Y^{(K)})$. To be more concrete, we present a few example machine learning problems and let them run through our development.

**Example: Translation Problem.** In this problem, the input $X$ is a sentence, or a sequence of words, in the source language, and output $Y$ is the sequence of words in the target sentence, where $Y^{(k)}$ denotes the $k^{\text{th}}$ word.

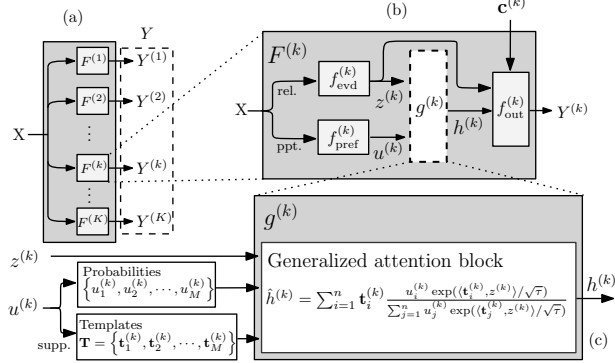

Figure 1: A conceptual graph of the deep learning model that we work with. The block $g^{(k)}$ is the one we will investigate.

**Example: Image Captioning.** In this problem, the input $X$ is a raw image and output $Y$ is the sequence of words in the caption, where $Y^{(k)}$ denotes the $k^{\text{th}}$ word.

**Example: Filling in the blanks task.** This task has been used to train the BERT model [5]. The input $X$ is a sequence of words with certain percentage of words masked. The output $Y$ are the predicted masked words, where $Y^{(k)}$ denotes the $k^{\text{th}}$ masked one.

The objective of any of these problems and that we address in this paper is to learn a function $F$, mapping from the space of $X$ to the space of $Y$ so that $Y = F(X)$. We will denote by $F^{(k)}$ the part of $F$ responsible for predicting $Y^{(k)}$ (Fig 1a), namely, $Y^{(k)} = F^{(k)}(X)$. Although we here express $F$ as separate functions $(F^{(1)}, F^{(2)}, \ldots, F^{(K)})$, we note that it is in fact possible that different $F^{(k)}$'s share some component in common. We now focus on the design of $F^{(k)}$.

We restrict the architecture of $F^{(k)}$ to the form in Fig 1b with the main focus on the inference of $h^{(k)}$. The extraction of feature $h^{(k)}$ is via two parallel modules $f_{\text{evd}}^{(k)}$ and $f_{\text{pref}}^{(k)}$ that directly operate on the input $X$ followed by a function $g^{(k)}$ (in Fig 1c), which we will design.

**The Design Problem** We describe the problem of designing $g$ as follows.

Suppose that there is an unknown distribution $p^{(k)}$ on $\mathbb{R}^d$ whose mean vector is $h^{(k)}$, namely,

$$h^{(k)} = \int_{\mathbb{R}^d} \mathbf{a} p^{(k)}(\mathbf{a}) \, \mathbf{da}. \tag{4}$$

Let $u^{(k)}$ be another distribution on $\mathbb{R}^d$ that is generated as the output of a network module $f_{\text{pref}}^{(k)}$. Here $u^{(k)}$ is referred to as the preference distribution, which serves as a prior guess of $p^{(k)}$. Specifically $u^{(k)}$ puts non-zero probability masses on $M$ "template" vectors $\mathbf{t}_1^{(k)}, \mathbf{t}_2^{(k)}, \ldots, \mathbf{t}_M^{(k)}$ in $\mathbb{R}^d$, and their probabilities are respectively $u_1^{(k)}, u_2^{(k)}, \ldots, u_M^{(k)}$ (which sum to 1). Collectively, we will denote the set $\{\mathbf{t}_1^{(k)}, \mathbf{t}_2^{(k)}, \ldots, \mathbf{t}_M^{(k)}\}$ of templates by $\mathbf{T}^{(k)}$.

The preference distribution $u^{(k)}$ is considered as a good approximation of $p^{(k)}$, in the sense that the support of $p^{(k)}$ is contained in the set $\mathbf{T}^{(k)}$ of templates. Note that if $\mathbb{R}^d$ is the word embedding space

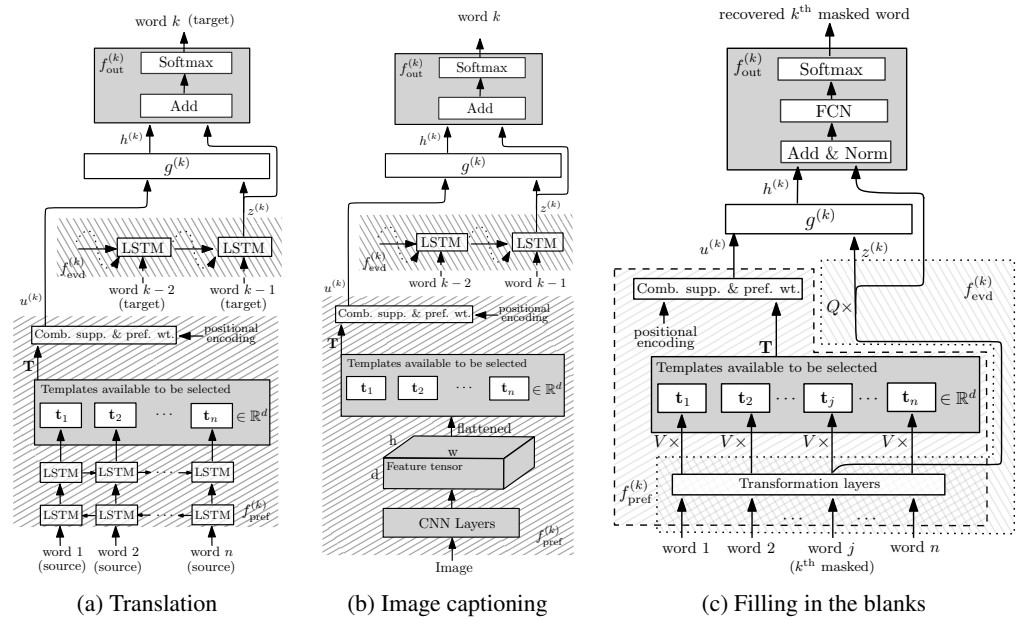

(a) Translation      (b) Image captioning      (c) Filling in the blanks

Figure 2: The model architectures of the three running examples. For the $f_{\text{evd}}^{(k)}$ in (a) and (b), the dashed links exist throughout the training and are replaced by the dotted ones in the generation stage.

for a large vocabulary, and if the size $M$ of the template set $\mathbf{T}^{(k)}$ is relative small, then restricting the support of $p^{(k)}$ to within $\mathbf{T}^{(k)}$ imposes a strong constraint on $p^{(k)}$.

On the other hand, $u^{(k)}$ is not a sufficiently accurate approximation of $p^{(k)}$, in the sense that $u^{(k)}$ may assign probabilities to $\mathbf{T}^{(k)}$ somewhat differently. Such inaccuracy shifts the mean $\mu^{(k)}$ of $u^{(k)}$ from the mean $h^{(k)}$ of $p^{(k)}$. Suppose that there is another piece of information $z^{(k)} \in \mathbb{R}^d$ that is generated by another network module $f_{\text{evd}}^{(k)}$ and provides information regarding the mean shift. In particular, we assume that $z^{(k)}$ is a noisy version of the shift, more precisely,

$$z^{(k)} = h^{(k)} - \mu^{(k)} + \epsilon, \tag{5}$$

where $\epsilon \sim \mathcal{N}(\mathbf{0}, \sigma^2 \mathbf{I})$ is the spherical Gaussian noise in $\mathbb{R}^d$ with covariance $\sigma^2 \mathbf{I}$. We refer to $z^{(k)}$ as the evidence.

Then the design problem is *to construct a function, or a network block, g, which infers the unknown distribution $p^{(k)}$ and hence its mean $h^{(k)}$ based on the evidence $z^{(k)}$ and the preference distribution $u^{(k)}$.*

This formulation of the design problem might seem peculiar at the first glance, but we will show via examples (see Fig 2) that such a problem naturally arises in the construction of many machine learning models in practice.

**Example: Translation Problem.** For the translation problem, consider the model implementation plotted in Fig 2a that is similar to the one proposed in [1]. We will focus on the part of the model responsible for inferring the $k^{\text{th}}$ word of the target sentence. In this model, $h^{(k)}$ corresponds to the constructed feature according to (4) that serves as an estimate of the context vector collecting the source sentence information. The estimated $h^{(k)}$ is then fed into a classifier $f_{\text{out}}^{(k)}$ to predict the $k^{\text{th}}$ word. The preference distribution $u^{(k)}$ is generated by $f_{\text{pref}}^{(k)}$ which takes the source sentence words as inputs. In particular, the support of $u^{(k)}$ consists of the source sentence word embeddings $\mathbf{T}$ (called annotations in [1]) which are pre-processed by two LSTM layers.[1] The preference weight for each template depends on some positional encoding functions, which, in principle, should assign higher

---

[1]In this model, given input $X$, all $u^{(k)}$'s share the same support $\mathbf{T}$. The superscripts of the templates are then omitted to show their independence from $k$. Similar comments apply to implementations of the other two running examples.

weights to the templates appearing in the similar locations to the words we are inferring (that is, $h^{(k)}$ is assumed to rely on the templates near $\mathbf{t}_k$ more heavily).

Note that the inferred $p^{(k)}$'s support must be a subset of $u^{(k)}$'s as it is reasonable to assume that the target sentence words only depend on those appearing in the source sentence. Besides, although the preference weights specified by the positional encoding functions could provide some *a priori* information for the templates' weights in $p^{(k)}$, they cannot be accurate as their inferences do not consider the previously generated words $Y^{(i<t)}$. This results in the mean $\mu^{(k)}$ shifted from $h^{(k)}$, which is estimated by $z^{(k)} = f_{\text{evd}}^{(k)}$ that takes all the previously generated words $Y^{(i<t)}$ into account using another LSTM layer. Thus, $h^{(k)}$ and $p^{(k)}$ should not be far from $z^{(k)} + \mu^{(k)}$ and $u^{(k)}$, respectively.

**Example: Image Captioning.** The caption generation model presented in Fig 2b has a similar architecture reported in [29]. This model shares the designs of $f_{\text{evd}}^{(k)}$ and $f_{\text{out}}^{(k)}$ with the translation model while $f_{\text{pref}}^{(k)}$ instead extracts the templates from a raw image using a CNN network. In general, a word's position in the caption is independent of the location of the object it describes in the image. Therefore, in this model, all templates extracted by the CNN share the same preference weight.

As similar objects appear in an image would have similar features extracted by the CNN (for example, a zebra and a horse), allowing similar templates not in $\mathbf{T}$ to participate in $h^{(k)}$'s estimation would possibly mix in information not contained in the raw image and harm the word inference accuracy. Therefore, we could improve the estimate of $h^{(k)}$ by choosing $p^{(k)}$ similar to $u^{(k)}$ in the sense that $p^{(k)}$'s support cannot contain elements not in $u^{(k)}$'s.

Intuitively, as the generation process proceeds, the context $h^{(k)}$ should be updated to provide relevant information in the image to facilitate the next word inference. Such change is governed by the caption's semantic evolution, which is captured by $z^{(k)} = f_{\text{evd}}^{(k)}$ that predicts the shift of the mean $\mu^{(k)}$ from $h^{(k)}$. For this reason, $\mu^{(k)} + z^{(k)}$ serves as an estimate of $h^{(k)}$ and should not be far away from it. Likewise, $u^k$ should be close to $p^{(k)}$.

**Example: Filling in the blanks task**. For the filling-in-the-blank tasks, let us consider a model architecture plotted in Fig 2c that is similar to the one used in BERT [5]. We focus on the inference of the $k^{\text{th}}$ masked word, which is assumed to be the $j^{th}$ word of the input sentence. In this model, $f_{\text{pref}}^{(k)}$ and $f_{\text{evd}}^{(k)}$ share the transformation layers (TL) that are commonly used in the NLP tasks to map one sequence of vector representations to another of the same length.[2] Taking the output sequence, $f_{\text{pref}}^{(k)}$ applies a linear map $V$ to each of its elements to form $\mathbf{T}$ as the support of $u^{(k)}$ while the preference weights are specified by some positional encoding functions. At the same time, $z^{(k)} = f_{\text{evd}}^{(k)}$ estimates $h^{(k)}$'s shift from the mean $\mu^{(k)}$ due to the variation of the local information. For the same reasons discussed in the previous two examples, we need $\mu^{(k)} + z^{(k)}$ close to $h^{(k)}$ while $p^{(k)}$ is close to $u^{(k)}$.

Notably the formulation of the problem is based on the assumption that the network modules $f_{\text{evd}}^{(k)}$ and $f_{\text{pref}}^{(k)}$ are fixed and generate $z^{(k)}$ and $u^{(k)}$ satisfying the above assumed properties. In reality, $f_{\text{evd}}^{(k)}$ and $f_{\text{pref}}^{(k)}$ are in fact obtained via training. However, we argue that if $g$ is made to satisfy our design objective, then we can at least *interpret* $f_{\text{evd}}^{(k)}$ and $f_{\text{pref}}^{(k)}$ obtained from training as serving to produce $z^{(k)}$ and $u^{(k)}$ with our desired properties.

## 5   Formulation of an Optimization Problem

The discussion made in the previous section implies that the key optimization problem we are about to focus on should ensure

1. $h^{(k)}$ is not too far from $\mu^{(k)} + z^{(k)}$, where $h^{(k)}$ is constructed by $p^{(k)}$ according to (4) and $\mu^{(k)}$ is the mean of the preference distribution $u^{(k)}$.

2. $p^{(k)}$ is close to $u^{(k)}$ while $p^{(k)}$'s support must be a subset of $u^{(k)}$'s.

---

[2]Typical implementation of such layers include convolution layers, recurrent layers and self-attention layers.

These two desiderata prompt us to optimize:

$$\min_p \frac{\alpha}{2} \left\| \left(\mu^{(k)} + z^{(k)}\right) - \int_{\mathbb{R}^d} \mathbf{a} p(\mathbf{a}) \, d\mathbf{a} \right\|^2 + \mathcal{K}(p, u^{(k)}) \tag{6}$$

with $\alpha > 0$ responsible for the relative strength of the two terms, where $\mathcal{K}(p, u^{(k)})$ denotes the KL divergence from $p$ to $u^{(k)}$. Remarkably, $\mathcal{K}(p, u^{(k)})$ has a finite value if and only if $p^{(k)}$ has non-zero values on the support of $u^{(k)}$. Thus, both requirements in the second desideratum are satisfied by using the KL divergence as a measure for the closeness of $p^{(k)}$ and $u^{(k)}$. Let $\tilde{p}^{(k)}$ be the minimizer of (6). The estimate of $h^{(k)}$ is

$$\hat{h}^{(k)} = \int_{\mathbb{R}^d} \mathbf{a} \tilde{p}^{(k)}(\mathbf{a}) \, d\mathbf{a}. \tag{7}$$

Naturally, this optimization problem can be derived from two different, though, related perspectives.[3]

**A maximum likelihood perspective.** The optimization problem in (6) can be derived using the maximum log likelihood method by treating the KL-divergence term as a regularizer. According to (5), the difference $(\mu^{(k)} + z^{(k)}) - h^{(k)}$ follows a Gaussian distribution $\mathcal{N}(\mathbf{0}, \sigma^2 \mathbf{I})$. This implies the log likelihood function $\ell(z^{(k)}) \propto -\frac{1}{2\sigma^2} \left\| (\mu^{(k)} + z^{(k)}) - h^{(k)} \right\|^2$. Maximizing it with the KL-divergence term as a regularizer is the same as minimizing

$$\frac{1}{2\sigma^2} \left\| \left(\mu^{(k)} + z^{(k)}\right) - h^{(k)} \right\|^2 + \eta \mathcal{K}(p, u^{(k)}), \tag{8}$$

where $\eta > 0$ controls the strength of the regularization. Substituting (4) into (8) followed by rearrangement yields

$$\min_p \frac{1}{2\eta\sigma^2} \left\| \left(\mu^{(k)} + z^{(k)}\right) - \int_{\mathbb{R}^d} \mathbf{a} p(\mathbf{a}) \, d\mathbf{a} \right\|^2 + \mathcal{K}(p, u^{(k)}), \tag{9}$$

which is equivalent to (6) by setting $\alpha^{-1} = \eta\sigma^2$.

**A maximum entropy on the mean perspective** Consider a problem that seeks a distribution $p$ such that the expectation $\int_{\mathbb{R}^d} \mathbf{a} p(\mathbf{a}) \, d\mathbf{a}$ is not far from $\mu^{(k)} + z^{(k)}$. In particular, we require

$$\left\| \left(\mu^{(k)} + z^{(k)}\right) - \int_{\mathbb{R}^d} \mathbf{a} p(\mathbf{a}) \, d\mathbf{a} \right\|^2 \leq \frac{1}{2\alpha}. \tag{10}$$

Note that, given $z^{(k)}$, there are infinitely many $p$'s that satisfy the constraints, which makes it difficult to pick a "best" $p$ for later use. A technique known in information theory as the maximum entropy on the mean (MEM) [6, 21] solves this problem by picking the best guess of the ground truth $p^*$ that simultaneously satisfies (10) and minimizes the KL divergence to the preference distribution $u^{(k)}$. That is,

$$\tilde{p}^{(k)} = \arg\min_p \mathcal{K}(p, u^{(k)}) \qquad \text{subject to} \qquad \left\| \left(\mu^{(k)} + z^{(k)}\right) - \int_{\mathbb{R}^d} \mathbf{a} p(\mathbf{a}) \, d\mathbf{a} \right\|^2 \leq \frac{1}{2\alpha}, \tag{11}$$

which is also the minimizer of (6) according to Equation (18) of [21] and Corollary 4.9 of [2].

## 6  Optimal Solution

Rioux et al. proved that the optimization problem stated in (6) has the following Fenchel dual (see Theorem 2 of [21]):

**Theorem 1.** *The dual of (6) is given by*

$$\max_{\lambda \in \mathbb{R}^d} \left\{ \left\langle \lambda, \mu^{(k)} + z^{(k)} \right\rangle - \frac{1}{2\alpha} \|\lambda\|^2 - \log \int_{\mathbb{R}^d} u^{(k)}(\mathbf{a}) \exp\langle \mathbf{a}, \lambda \rangle \, d\mathbf{a} \right\}. \tag{12}$$

*Given a maximizer $\lambda^*$ of (12), one can recover the minimizer $\tilde{p}^{(k)}$ of (6) via*

$$\tilde{p}^{(k)}(\mathbf{a}) = \frac{u(\mathbf{a}) \exp\langle \mathbf{a}, \lambda^* \rangle}{\int_{\mathbb{R}^d} u(\mathbf{a}') \exp\langle \mathbf{a}', \lambda^* \rangle \, d\mathbf{a}'}. \tag{13}$$

---

[3]In fact, there is also a Bayesian perspective to derive the problem, which will be discussed in Appendix A.

By Theorem 1, the estimated $h^{(k)}$ defined in (7) can be re-written as

$$\hat{h}^{(k)} = \int_{\mathbb{R}^d} \mathbf{a}\, \tilde{p}^{(k)}(\mathbf{a})\, d\mathbf{a} = \int_{\mathbb{R}^d} \mathbf{a}\, \frac{u^{(k)}(\mathbf{a}) \exp\langle \mathbf{a}, \lambda^* \rangle}{\int_{\mathbb{R}^d} u^{(k)}(\mathbf{a}') \exp\langle \mathbf{a}', \lambda^* \rangle\, d\mathbf{a}'}\, d\mathbf{a}, \tag{14}$$

where $\lambda^*$ is a maximizer of (12).

In general, $\lambda^*$ does not have a closed-form expression in terms of $\alpha$, $u^{(k)}$ and $z^{(k)}$, and a standard paradigm is to search for it using gradient ascent-based methods. In this paper, we will not search for $\lambda^*$ in this way; instead, we will derive a closed-form expression to approximate it. Remarkably, this takes the form of the generalized attention presented in Fig 1.

Note that the integration in (12) equals $\mathbb{E}_{u^{(k)}}[\exp\langle W, \lambda \rangle]$, the expectation of the random variable $\exp\langle W, \lambda \rangle$ where $W$ has the probability distribution $u^{(k)}$. The expectation is just the moment generating function (MGF) $M(\lambda)$ of $W$, and the value $\log M(\lambda)$ is called the cumulant of $W$ [17, p.26], which has an expansion [17, (2.4)]

$$\log M(\lambda) = \left\langle \mu^{(k)}, \lambda \right\rangle + \frac{1}{2} \left\langle \lambda, \Sigma^{(k)} \lambda \right\rangle + \mathcal{O}(\|\lambda\|^3), \tag{15}$$

where $\mu^{(k)} = \int_{\mathbb{R}^d} \mathbf{a}\, u^{(k)}(\mathbf{a})\, d\mathbf{a}$ and $\Sigma^{(k)} = \int_{\mathbb{R}^d} \left( \mathbf{a} - \mu^{(k)} \right) \left( \mathbf{a} - \mu^{(k)} \right)^T u^{(k)}(\mathbf{a})\, d\mathbf{a}$ respectively denote the expectation and the variance-covariance matrix of $W$.

Now we assume that $\alpha$ is small and we argue that this assumption is justified in practice. For instance, in the translation task, all of words in the dictionary can serve as candidate templates, which could be more than 10,000, but $u^{(k)}$ reduces this size to the length of the source sentence (usually less than tens of words). The inference of $p^{(k)}$ should strongly anchor around this prior information; consequently the information provided by $z^{(k)}$ should weigh less. On the other hand, $z^{(k)}$ can hardly provide an accurate estimate of the mean shift, since the generation of $z^{(k)}$ is often ignorant of the templates selected by $u^{(k)}$ (for example, in the example translation and image captioning models) or generated by a low-capacity module (as in the example filling-in-the-blank model). For these reasons, one should de-emphasize the constraint imposed by $z^{(k)}$ and hence choose a small $\alpha$.

When $\alpha$ is picked to be small enough (see (12)), the optimization of $\lambda$ gets a large penalty on its L2 norm and thus, $\|\lambda^*\|$ is close to zero. Then, by (15), we have

$$\log \int_{\mathbb{R}^d} u^{(k)}(\mathbf{a}) \exp\langle \mathbf{a}, \lambda^* \rangle\, d\mathbf{a} = \log M(\lambda^*) \approx \langle \mu^{(k)}, \lambda^* \rangle + \frac{1}{2} \langle \lambda^*, \Sigma^{(k)} \lambda^* \rangle. \tag{16}$$

Substituting (16) into (12) followed by setting the derivative with respect to $\lambda$ to zero yields

$$\lambda^* = \alpha (I_d + \alpha \Sigma^{(k)})^{-1} z^{(k)}, \tag{17}$$

where $I_d$ denotes the $d \times d$ identity matrix. As $\alpha$ is assumed close to zero, (17) is further reduced to

$$\lambda^* = \alpha z^{(k)}. \tag{18}$$

Plugging the expression into (14) gives the result stated as follows:

**Theorem 2.** *For a small enough $\alpha > 0$, the estimated $h^{(k)}$ defined in (7) can be approximated by*

$$\hat{h}^{(k)} = \int_{\mathbb{R}^d} \mathbf{a}\, \frac{u^{(k)}(\mathbf{a}) \exp(\alpha \langle \mathbf{a}, z^{(k)} \rangle)}{\int_{\mathbb{R}^d} u^{(k)}(\mathbf{a}') \exp(\alpha \langle \mathbf{a}', z^{(k)} \rangle)\, d\mathbf{a}'}\, d\mathbf{a}. \tag{19}$$

For the case that $u^{(k)}$ is a discrete distribution with support $\{\mathbf{t}_1^{(k)}, \mathbf{t}_2^{(k)}, \ldots, \mathbf{t}_n^{(k)}\}$ and the preference probability $\{u_1^{(k)}, u_2^{(k)}, \ldots, u_n^{(k)}\}$, (19) becomes simply

$$\hat{h}^{(k)} = \sum_{i=1}^n \mathbf{t}_i\, \frac{u_i^{(k)} \exp(\alpha \langle \mathbf{t}_i, z^{(k)} \rangle)}{\sum_{j=1}^n u_j^{(k)} \exp(\alpha \langle \mathbf{t}_j, z^{(k)} \rangle)}. \tag{20}$$

In Fig 3, we set $d = 2$ and visualize the approximation of $h^{(k)}$ for different selections of $\alpha$. We can observe that, as $\alpha$ decreases, (20) outputs a better approximation of $\hat{h}^{(k)}$.

Let $\alpha = \tau^{-\frac{1}{2}}$, we rewrite Theorem 2 as follows for later reference.

**Corollary 1.** *For a sufficiently large $\tau$, the best guess of $h^{(k)}$ defined in (7) with $\alpha = \tau^{-\frac{1}{2}}$ equals*

$$\hat{h}^{(k)} = \sum_{i=1}^n \mathbf{t}_i^{(k)}\, \frac{u_i^{(k)} \exp(\langle \mathbf{t}_i^{(k)}, z^{(k)} \rangle / \sqrt{\tau})}{\sum_{j=1}^n u_j^{(k)} \exp(\langle \mathbf{t}_j^{(k)}, z^{(k)} \rangle / \sqrt{\tau})}. \tag{21}$$

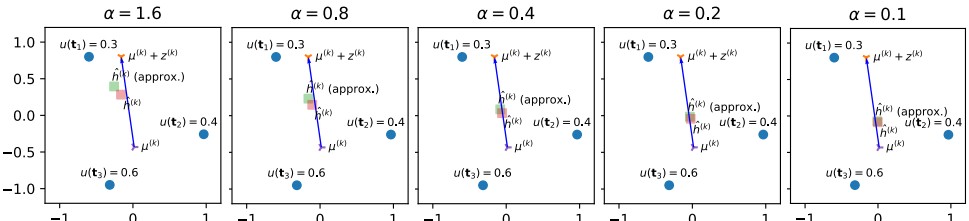

Figure 3: The approximation of $\hat{h}^{(k)}$ for different choices of $\alpha$. The dots in light blue compose the support of discrete $u^{(k)}$ with the preference weights labelled above. The dark blue arrow starting from the mean $\mu^{(k)}$ of $u^{(k)}$ denotes the evidence $z^{(k)}$. The red square marks the $\hat{h}^{(k)}$ constructed by (14) with the $\lambda^*$ maximizes (12), while the green one marks the $\hat{h}^{(k)}$ approximated by (20). As we can observe, (20) gives a precise approximation of $\hat{h}^{(k)}$ when $\alpha$ is sufficiently small.

## 7    Discussion

In Section 6, we derived an alternative expression of $\hat{h}^{(k)}$ defined in (7) by solving the Fenchel dual of the optimization problem stated in (6). Although the expression is not in closed form, as we are only interested in the case when $\alpha$ is small, a closed-form approximation of $\hat{h}^{(k)}$ is derived in Theorem 2 and reduced to the form stated in (21) when considering a discrete distribution $u^{(k)}$.

As we pointed out, the block $g^{(k)}$ in Fig 2a, Fig 2b and Fig 2c is expected to find the inferred $\tilde{p}^{(k)}$ minimizing (6) followed by plugging it into (7) to construct $\hat{h}^{(k)}$. Therefore, one can complete the architecture designs of the three running examples by replacing $g^{(k)}$ with a network layer implementing (21), namely, the structure in Figure 1 (c).

**The relationship between the optimal solution and the attention models.** Remarkably, the expression stated in (21) gives a generalized attention block. By setting $u_i^{(k)} = \frac{1}{n}$ for all $i$, the expression is equivalent to the well known dot-product attention [15], which is also applied in the transformer network [25]. The equivalence of the expression of $\hat{h}^{(k)}$ and the dot-product attention layer tells us: (a) *by applying a dot-product attention layer in a model, we essentially ask the model to perform an optimization task defined in (6) and construct the output according to (7).* (b) *the derivation of $h^{(k)}$ depends on two relatively independent pieces of information: a preference distribution given the global information and an estimate of the output's deviation from the preference distribution's mean according to some local information. This suggests that the design of attention-based model can be decomposed into two parts that respectively estimate these two values.*

**The model consisting of a stack of attention layers.** Although our discussion focuses on the case that contains a single attention layer, any attention layer $\mathcal{L}$ in an attention stack fits our frameworks (see Fig 1). In particular, all the attention layers closer to the input $X$ than $\mathcal{L}$ can be grouped into the functions $f_{\text{pref}}^{(k)}$ or $f_{\text{evd}}^{(k)}$. For those layers that take the current layer's output as input, we can group them into $f_{\text{out}}^{(k)}$, where $\mathbf{c}^{(k)}$ may contain the outputs of other attention layers working in parallel.

**T5 transformer implicitly adopts the generalized attention structure.** We now show that T5 transformer [19] can be seen as a realization of the generalized attention in (21), where the preference weights $u^{(k)}$ unifies the concepts of word masks and T5's positional encoding functions. Consider the running example: filing in the blanks, with the preference distribution

$$u^{(k)}(\mathbf{t}_i) = \begin{cases} 0 & \text{if the } i^{\text{th}} \text{ word is masked} \\ \exp(b_{j-i})/Z & \text{otherwise,} \end{cases} \tag{22}$$

where $Z$ is a normalizing constant and $b_{j-i}$ is a trainable scalar that only depends on the relative position of word $i$ and word $j$ (which is the $k^{\text{th}}$ masked word that we are inferring). Substituting such $u^{(k)}$ into (21) with $\tau = d$ yields

$$\hat{h}^{(k)} = \sum_{i=1}^{n} \mathbf{t}_i \frac{\exp\left(\frac{\langle \mathbf{t}_i, z^{(k)} \rangle}{\sqrt{d}} + b_{j-i} + \mathbf{1}_{\text{masked}}(i)\right)}{\sum_{l=1}^{n} \exp\left(\frac{\langle \mathbf{t}_l, z^{(k)} \rangle}{\sqrt{d}} + b_{j-l} + \mathbf{1}_{\text{masked}}(l)\right)}, \tag{23}$$

where $\mathbf{1}_{\text{masked}}(i)$ is an indicator function that equals $-\infty$ if word $i$ is masked and zero otherwise. The expression in (23) has the same structure as that adopted in T5 transformer, where the indicator function serves as the mask function to prevent the model from assigning weights to the masked words. In this way, the concepts of word masks and the positional encoding functions are unified by $u^{(k)}$ in (22). Conversely, T5 transformer is a realization of the generalized attention with the preference weights $u^{(k)}$ specified in (22).

**Generalized attention structures suggested by the optimal solution.** While T5 transformer has implicitly adopted the generalized attention, (21) hints further generalizations could be made. For instance, in T5 transformer, the function outputting template's preference weights only considers the word masks and the word's relative positions. This function could be generalized to also consider the input sentence contexts, and the output weights encode the importance of each word before giving the local information stored in $z^{(k)}$. The same idea could be applied to the image captioning example to replace the uniform preference weights. By adding a neural network taking the input image to generate non-uniform preference weights, we devise a mechanism to estimate the importance of each part of the image before the caption generation. In this way, the newly added network collects global information from the image to propose a preference distribution, which could be updated locally based on current generation stage encoded in $z^{(k)}$.

Moreover, although we mainly focus on the case when $u^{(k)}$ is discrete, we want to emphasize that the analysis performed in Section 6 also covers continuous $u^{(k)}$. This hints that a continuous attention mechanism could also be implemented, which might prove to be useful in some applications.

**Limitations and other comments.** The approximations performed in (15) and (18) have implicitly assumed that random variable $W$ following distribution $u^{(k)}$ has bounded moments. For a discrete $u^{(k)}$ with fixed support $\mathbf{T} = \{\mathbf{t}_1, \mathbf{t}_2, \ldots \mathbf{t}_n\}$, all the moments are bounded and we can always pick a small enough $\alpha$ (or equivalently large enough scaling factor $\tau$ in Cor 1) to make the approximation meet our requirements. A concern may arise as the support $\mathbf{T}$ in our running examples are supplied by some neural layers, which could output templates of increasing norms as the training evolves. This problem could be alleviated by adding norm regularization or using normalized templates instead.

# 8 Conclusion

This paper presented a new perspective to understand the attention mechanism by showing that it can be treated as realizing a solver of a family of inference tasks. These tasks involve improving the noisy estimate of a distribution $p$'s mean by a preference distribution that encodes some beliefs of $p$'s value. We have used three running examples with the typical model architectures to show that such tasks naturally exist in neural network design. We then abstracted a convex optimization problem from these tasks and derived a closed-form approximation of the optimal solution by solving the problem's Fenchel dual. We find that the closed-form approximation can be seen as a generalized attention layer and show that one of its special cases is equivalent to the dot-product attention adopted in transformers. We further performed an analysis on the general form and showed that T5 transformer implicitly adopts the generalized attention structure with attention weights unifying the concepts of the word masks and the positional encoding functions.

This paper is the first work that presents a principled justification for the design of attention modules in neural networks. The generalized attention structure presented in this paper potentially opens a door to a wide design space. For example, the preference weights need not be derived from the positional encoding functions; they could integrate a variety of information provided by other components of the network. Additionally, this research might have pointed to new ways to analyze the functioning of a neural network component, namely, via isolating the component from the complex network structure and asking: is there a "local problem" that is solved by the design of this component?

**Potential negative societal impacts.** This paper presents a new perspective to understand attention and derived a generalized attention structure. Our work is foundational, which we believe does not have direct negative societal impacts. Due to the very wide range of applications of attention, such as self-driving [12] and healthcare [16], our work may have unexpected negative impacts on these areas.

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
