# A    A Bayesian perspective to derive (6)

477  Given observed data, Bayesian inference allows us to derive a distribution of the parameters of a
478  statistical model. By considering $\mu^{(k)} + z^{(k)}$ as the observed data and $p$ as a model parameter, we
479  will show that picking the $p$ that minimizes (6) is the same as choosing the $p$ that has the largest
480  probability in the derived distribution. In (5), we have assumed that $(\mu^{(k)} + z^{(k)}) - h^{(k)}$ follows a
481  spherical Gaussian distribution $\mathcal{N}(\mathbf{0}, \sigma^2\mathbf{I})$, where $h^{(k)}$ is the mean of $p$. Therefore, given $p$, we also
482  have

$$\Pr(\mu^{(k)} + z^{(k)}|p) = \Pr(\mu^{(k)} + z^{(k)}|h^{(k)}) \propto \exp\left(-\frac{1}{2\sigma^2}\left\|(\mu^{(k)} + z^{(k)}) - h^{(k)}\right\|^2\right). \quad (24)$$

483  Here, we let the prior distribution of $p$ satisfy

$$\Pr(p|u^{(k)}) \propto \exp\left(-\eta\mathcal{K}(p, u^{(k)})\right), \quad (25)$$

484  where $\eta > 0$ is a super parameter that controls the probability decreasing speed as $p$ deviates
485  from $u^{(k)}$. Then the posterior distribution of $p$ satisfies

$$\Pr(p|\mu^{(k)} + z^{(k)}, u^{(k)}) \propto \Pr(\mu^{(k)} + z^{(k)}|p)\ \Pr(p|u^{(k)})$$
$$\propto \exp\left(-\frac{1}{2\sigma^2}\left\|(\mu^{(k)} + z^{(k)}) - h^{(k)}\right\|^2 - \eta\mathcal{K}(p, u^{(k)})\right).$$

486  Finding $p^*$ that maximizes $\Pr(p|\mu^{(k)} + z^{(k)}, u^{(k)})$ is the same as finding

$$p^* = \arg\min_p \left\{\frac{1}{2\sigma^2}\left\|(\mu^{(k)} + z^{(k)}) - h^{(k)}\right\|^2 + \eta\mathcal{K}(p, u^{(k)})\right\}$$
$$= \arg\min_p \left\{\frac{1}{2\eta\sigma^2}\left\|(\mu^{(k)} + z^{(k)}) - h^{(k)}\right\| + \mathcal{K}(p, u^{(k)})\right\},$$

487  which is equivalent to (6) by setting $\alpha^{-1} = \eta\sigma^2$.