# OpenReview forum: "Attention as Inference via Fenchel Duality"
_NeurIPS.cc/2021/Conference — NeurIPS 2021 Submitted_

### Official Review · Reviewer_gF3T · 2021-07-06

**Rating:** 7
**Confidence:** 3

**Summary:**

This paper provides a new perspective on the attention mechanism. More precisely, the paper claims that an attention module can be seen as estimating the distribution solving a specific optimization problem. This problem follows the assumption that many deep learning models use a module updating a global guess with local information. The problem consists in minimizing both the discrepancy between an adjusted value of the mean and the mean of the estimated distribution, and between a prior distribution and the estimated distribution. Using some approximation, the solution corresponds to a generalized form of the dot-product attention.

**Limitations And Societal Impact:**

See above (Cons) for the limitations of the paper from my point of view.
I think the authors correctly address the potential negative societal impact.

**Main Review:**

Although the perspective provided seems at first glance artificial, it is original, sound, and seems indeed related to attention. The paper is globally well-written. To my mind, some points (how does this perspective apply in practice, some points to clarify) are worth being adressed before recommending acceptance.

Pros:
- To the best of my knowledge, the paper indeed provides a new, nice perspective on attention.
- The paper provides insightful links between their formulation of attention and classical deep learning architectures.

Cons:
- The perspective proposed by the paper is potentially insightful but the authors does not bring out whether their perspective really applies in practice: would real, learned attention mechanisms indeed provide a good approximation to the optimization problem (6)?
- Conversely, the paper could benefit from an example where solving the optimization problem can effectively replace a learned attention mechanism.
- It is not clear to me whether this perspective really applies in transformers (see below). This may be due to a misunderstanding from my side though.

Questions and remarks:
- l21-22: what is ``"poorly understood" in attention?
- Section 2: There are works related to attention that could be worth citing in Section $2$, see non-local means [1], and attention as a kernel smoothing [2] which also proposes a generalized attention mechanism.
- Section 4: previous work learn a template from the whole data set for computing the attention weights in linear time, e.g, [3][4]: would this improve the estimation of $p^{(k)}$ in your perspective?
- l114: early explanation/intuition on $f_{prev}$ and $f_{evd}$ could be useful.
- l120: inconsistency in fonts for the Real symbol.
- l134:  could you elaborate on this problem? Why would we want to solve it in the context of deep learning?
- l174: in BERT, the attention weights are computed with keys and queries matrices, and not with values and queries as suggested by Figure 2c. Moreover, the attention output is added to the input feature map which is different from what Figure 2c seems to suggest. What is your view on these two potential differences?
- Figure 3: if $\alpha$ is small, it seems that the estimate of $\hat{h}^{(k)}$ goes away from $\mu^{(k)} + z^{(k)}$, making the evidence term $z^{(k)}$ potentially useless?

---------

[1] Buades et al. : Non-local Means Denoising, IPOL 2011

[2] Tsai et al. : Transformer Dissection: A Unified Understanding of Transformer's Attention via the Lens of Kernel, EMNLP 2019

[3] Mialon et al. : A Trainable Optimal Transport Embedding for Feature Aggregation and its Relationship to Attention, ICLR 2021

[4] Kolouri et al. : Wasserstein Embedding for Graph Learning, ICLR 2021

**Time Spent Reviewing:**

5

---

> ### Author Response · Authors · 2021-08-10
> **Clarification**
>
> Thank you very much for your insightful comments. Our responses follow.
>
>
> - The perspective proposed by the paper is potentially insightful but the authors does not bring out whether their perspective really applies in practice: would real, learned attention mechanisms indeed provide a good approximation to the optimization problem (6)? Conversely, the paper could benefit from an example where solving the optimization problem can effectively replace a learned attention mechanism.
>
> **Response**.  This is nice advice. We have provided extra evidence to support our work. Please refer to our comment box entitled "Extra experimental results for reviewers" for details.
>
> - l21-22: what is "poorly understood" in attention?
>
> **Response**. What we mean here is that the design principle of attention has not been well studied in the literature and that there is no indepth understanding as to why attention-based models (e.g. BERT) have significantly better performance than other models. We will update our sentence in the revised version to avoid any potential confusion.
>
>
> - Section 2: There are works related to attention that could be worth citing in Section 2, see non-local means [1], and attention as a kernel smoothing [2] which also proposes a generalized attention mechanism.
>
> **Response**. Thank you for your pointers. We will add these references in the revised version.
>
>
> - Section 4: previous work learns a template from the whole data set for computing the attention weights in linear time, e.g, [3][4]: would this improve the estimation of $p^{(k)}$ in your perspective?
>
> **Response**. Yes, this will help a lot for the optimal-transport-based version of our framework (see below), although for the current KL-based version, the involved computation is already very light-weight.
>
> We reiterate a comment also made to Reviewer p7xB: our interpretation of the attention algorithm suggests the existence of more general and powerful attention structures on the vocabulary level, and we are currently polishing our results for a follow-up paper. In more detail, as mentioned in the present paper, the KL-divergence in Eq (6) forces the distribution to be estimated to share its support with the preference distribution. This property may not be desired in some tasks where the attention structures are broadly applied. For example, in a translation task, the target sentence is unchanged if we replace some words in the source sentence with their synonyms (like replacing "good" with "nice"). Such a property of the data is not accounted for in the design of current attention because the KL-divergence forces models to put zero weights on the words not in the source sentence. This problem is discussed and solved in our follow-up paper by replacing the KL divergence with an optimal transport-based measure, where the words "similar" to the ones in the source sentence will also be attended. The technique introduced in [4] is likely to help reduce the computation complexity for that purpose. Thank you for pointing us to this work!
>
> - l114: early explanation/intuition on $f_{pref}$ and $f_{evd}$ could be useful.
>
> **Response**. We will revise our paper to give more intuitive explanations on these two functions when they are first defined.
>
> - inconsistency in fonts for the Real symbol.
>
> **Response**. We will fix this typo in the revised version.
>
>
> - l134: could you elaborate on this problem? Why would we want to solve it in the context of deep learning?
>
> **Response**. The overall philosophy of this work is to first isolate the attention module from a deep neural network and ask the question: what problem is this module solving? In this paper, we show that the module can be formulated as solving a specific inference problem. This provides insight into design choices in the attention module.
>
> In a nutshell, the inference problem is of the following nature: suppose we have a noisy observation of some distribution's mean and we also have a rough prior knowledge of how this distribution looks. The problem is to infer the distribution's mean using both these ingredients. In this paper, we do this by first using the noisy mean to improve the estimation of the distribution, and then taking its mean as our final estimation.
>
> To see this in action in a concrete deep learning problem, let us consider a translation task and suppose the model is trying to infer the $k^{\rm th}$ word in the target sentence. Based on the previously generated $k-1$ words, the model can have some preliminary idea on what the $k^{\rm th}$ word should be (e.g. a word related to animals or a word related to weather, etc.). This information could be encoded in the observation of mean, which could be very noisy (as there are many different animals or different weathers). On the other hand, the prior knowledge of the distribution gives the model some idea about the location in the source sentence that contains information for inferring the $k^{\rm th}$ word. Combining these two sources of information, the model can find the exact words in the source sentence to attend (corresponding to the improved estimation of the distribution) and then use the mean of their embeddings to infer the $k^{th}$ word.
>
>
> - l174: in BERT, the attention weights are computed with keys and queries matrices, and not with values and queries as suggested by Figure 2c. Moreover, the attention output is added to the input feature map which is different from what Figure 2c seems to suggest. What is your view on these two potential differences?
>
> **Response**. The example in Figure 2c should be seen as a special case of the regular transformer, specifically, the case where the key matrix $K$ and value matrix $V$ are tied. Note that this special transformer structure can be reparametrized to give a general one, as has been shown by Ramsauer et al. in (https://openreview.net/forum?id=tL89RnzIiCd). In particular, let $Y\in \mathbb{R}^{n\times d}$ denote the input of the special transformer, where $n$ is the number of words and $d$ is the word's dimension. Then in the example presented by Figure 2c, we have the attention structure like
>
> $$VY {\rm softmax} (\alpha (VY)^T (QY))$$
>
> where $V, Q \in \mathbb{R}^{d\times d}$, $\alpha$ is the one defined in Eq (6),  and softmax applies to the column vectors. Write $V = V' K'$ for some $V', K' \in \mathbb{R}^{d\times d}$. Then we have
>
> $$V' K' Y {\rm softmax} (\alpha (V' K' Y)^T (QY)).$$
>
> That is,
>
> $$(V' K') Y {\rm softmax} (\alpha Y^T K'^T (V'^T Q) Y).$$
>
> Let $Q' = V'^TQ$ and $V''=V'K'$. We get
>
> $$V'' Y {\rm softmax} (\alpha (K' Y)^T (Q' Y)),$$
>
> which is the regular transformer structure.
>
> For your second question, this is a typo. The skip connection should not be transformed by $Q$. We will fix this typo in the revised version.
>
> - Figure 3: if $\alpha$ is small, it seems that the estimate of $\hat{h}^{(k)}$ goes away from $\mu^{(k)} + z^{(k)}$, making the evidence term  $z^{(k)}$ potentially useless?
>
> **Response**. If $\alpha = 0$, then $z^{(k)}$ is useless. However, we want to note that we do not need $\alpha$ arbitrarily close to zero to get a good approximation. For example, in Figure 3, we can observe that the approximated value almost coincides with the exact one when $\alpha = 0.1$ and $0.2$. And for these two selections, $\hat{h}^{(k)}$ still significantly deviates from $\mu^{(k)}$ (which corresponds to the case when $\alpha =0$ and $z^{(k)}$ is useless). Therefore, $z^{(k)}$ still largely affects the final estimation results (see Figure 3).

---

> > ### Comment · Reviewer_gF3T · 2021-09-01
> > **Thank you for your answer**
> >
> > I want to thank the authors for their detailed comment. I think they did a good job at answering my concerns and I am willing to increase my score from 5 to 7: the paper may lack some clarity but proposes an original perspective on attention as well as promising experimental results.

---

### Official Review · Reviewer_4PEc · 2021-07-11

**Rating:** 6
**Confidence:** 1

**Summary:**

The paper casts attention as a solver for an inference task that consists in estimating the mean of an unknown distribution given two information: an unreliable estimate and a prior distribution. The authors start by explaining this inference task and showing how this setup encompasses several real-life examples (translation, image captioning, filling the blanks). Then, they derive a convex optimization problem from this task and solve it by looking  at its Fenchel dual. Finally, the authors show that the solution obtained from this problem yields a generalized attention that is implicitly used in T5 Transformer.

**Limitations And Societal Impact:**

The authors dedicated a whole paragraph regarding the limitations of this paper which is definitely an advantage of the paper.

**Main Review:**

As mentioned to the AC, I would like to first signal that I am not familiar at all with attention and NLP tasks. Therefore, it is very hard for me to assess whether the model proposed in Section 4 by the authors is valid or not.

Originality: The paper is definitely original and looks at a question that raises a lot of interest. The techniques used are classic in convex optimization but they are used in a novel way. I am not aware of any work modelling attention as a solution of a convex optimization problem. The related has the merit to cite other works that mathematically model attention.

Quality: The paper seems technically sound on the optimization side. I also appreciate the fact that the authors bring the weaknesses and limitations of their approach. I would like to raise some points:

1) The problem formulation (6) assumes that p is in the support of u^k (otherwise the KL divergence is not well defined). However, how do you enforce this fact? It does not seem obvious to me that you are enforcing this constraint in (6).

2) Is it realistic in the case of language to assume that the noise in (5) is a Gaussian random variable? The whole analysis in Section 5 seems to rely on that.

3) What is the choice of the prior that you would make when applying your method in practice? For instance in a translation task?

4) Do you think that it is possible to numerically compare the Transformer T5 with a vanilla Transformer? How does your model differ from the vanilla attention?

Clarity: I think that the paper is very well written and I appreciate the efforts of the author to explain in detail their motivation (Section 3) and the setup they propose (Section 4) by taking concrete examples.

Significance: I cannot really assess the significance of the paper as I am not an expert in NLP. I however believe that the authors should do more experiments on real-life NLP tasks and compare it to a vanilla transformer. This would help to assess the validity of their model and maybe figure out some limitations of their approach.

**Time Spent Reviewing:**

4 hours

---

> ### Author Response · Authors · 2021-08-10
> **Clarification**
>
> Thank you very much for your comments. Our responses follow.
>
>
> - The problem formulation (6) assumes that p is in the support of u^k (otherwise the KL divergence is not well defined). However, how do you enforce this fact? It does not seem obvious to me that you are enforcing this constraint in (6).
>
> **Response**. The KL divergence from $p$ to $u^{(k)}$ takes the form
> $$\mathcal{K}(p, u^{(k)}) = \int_{\mathbb{R}^d}p(\mathbf{a})\log\frac{p(\mathbf{a})}{u^{(k)}(\mathbf{a})}{\rm d}\mathbf{a}.$$
> As we can see, if there exsits $\mathbf{a}\in \mathbb{R}^d$ such that $p(\mathbf{a}) > 0$ but $u^{(k)}(\mathbf{a}) = 0$, the term $p(\mathbf{a})\log\frac{p(\mathbf{a})}{u^{(k)}(\mathbf{a})}$ approaches to the infinity (or not well-undefined). Therefore, to have the KL-divergence finite, we must have $p(\mathbf{a}) = 0$ whenever $u^{(k)}(\mathbf{a}) =0$. In other words, the support of $p$ must be in the support of $u^{(k)}$.
>
> In this paper, we define the optimization problem (6) with the restriction that $p$ is in the support of $u$. We show in a few examples that this condition is indeed satisfied in several practical settings. However, as you are perhaps pointing to, such a restriction might limit the design of attention mechanisms. In our subsequent work (see also discussion with Reviewer p7xB), we relax this restriction by using an optimal-transport measure to replace the KL divergence.
>
> - Is it realistic in the case of language to assume that the noise in (5) is a Gaussian random variable? The whole analysis in Section 5 seems to rely on that.
>
> **Response**. Thanks for asking about this. Assuming the noise to be gaussian is certainly mathematically convenient. On the other hand, such an assumption can be justified by the central limit theorem (CLT). To see this, let us assume the error $\epsilon = z^{(k)}-(h^{(k)}-\mu^{(k)})$ in Eq (5) follows some zero-mean distribution with variance $\sigma^2$ (in case the mean is not zero, it can be absorbed in $z^{(k)}$). Now, suppose we have $n$ observations $\epsilon_1, \epsilon_2, \cdots, \epsilon_n$ on $\epsilon$. The CLT tells us the average $\bar{\epsilon} = \frac{1}
> {n}\sum_{i=1}^n \epsilon_i \rightarrow \mathcal{N}(0, \sigma^2/n)$ regardless of the distribution of $\epsilon$. Besides, we know that if $\epsilon_i\sim \mathcal{N}(0, \sigma^2)$, the average $\bar{\epsilon} \sim\mathcal{N}(0, \sigma^2/n)$ for all $n$. With these facts in mind, and considering that our model focuses on the fluctuation of the errors over all data samples, it is reasonable to  directly assume $\epsilon$ is Gaussian because, regardless, the limiting distrubtion of the average is unchanged.
>
> - What is the choice of the prior that you would make when applying your method in practice? For instance in a translation task?
>
> **Response**. The most natural and popular choice would be the categorical distribution. In particular, the distribution has a support consisting of finite templates
>
> $$\\{\mathbf{t}_1, \mathbf{t}_2, \cdots,  \mathbf{t}_n\\}.$$
>
> Each template $\bf t_i$ has a corresponding probability $ u_i \geq 0$ of being selected, and we have $\sum_{i=1}^n u_i =1$. In Section 7, we discussed what these $u_i$'s correspond to in the T5 transformer, which is a popular model architecture in today's natural language processing research.
>
>
> - Do you think that it is possible to numerically compare the Transformer T5 with a vanilla Transformer? How does your model differ from the vanilla attention?
>
> **Response**. Yes, this is possible. We want to note that the T5 transformer was first proposed by Raffel et al. (2020) in (https://arxiv.org/pdf/1910.10683.pdf) and is not our contribution. Our framework in fact can describe both the original (vanilla) transformer and T5. Although Raffel et al do not directly compare T5 with the vanilla transformer, they show that T5 achieves state-of-the-art results on many benchmarks covering summarization, question answering, text classification, etc. and, in particular, thus outperforms the vanilla transformer.

---

### Official Review · Reviewer_ZmjE · 2021-07-12

**Rating:** 5
**Confidence:** 4

**Summary:**

The paper introduces a novel interpretation of the attention algorithm and corresponding theoretical discussions. The paper suggests that the "(weighted) attention algorithm" can be interpreted as an approximate solution of a density estimation problem, where we have a noisy observation of a mean and a reference distribution (or "preference distribution" in the paper).

In the proposed interpretation, the density estimation solves a proximal-type optimization problem. The penalty function is a reverse KL to a reference distribution, and the objective function is an L2-norm between the model's mean and the observed mean. Here, the authors emphasize treating the observed mean as a sum of the reference's mean and a noisy observation of a mean-shift.
Since the optimization problem satisfies the strong duality, the optimization can be solved by its dual; however, it is still challenging to obtain its analytical solutions. Thus, in the dual problem, the authors introduce to approximate a term, a cumulant generating function of a reference distribution, by its second-order Taylor polynomial. The authors show that the closed-form solution of the resulting approximate problem is the dot-product attention, weighted by a reference distribution.

Furthermore, the authors show that the proposed interpretation includes a well-known T5 transformer as a special case. Assume we aim at solving the proposed optimization problem to estimate a density of a categorical distribution, where the distribution is on a set of M vectors (i.e., M keys). Given some context information, a set of networks predicts a preference distribution and a query, which is the mean-shift. The weight on the L2-norm in the optimization problem is the inverse of the temperature parameter. Then the T5 transformer's attention can be obtained by the expected value of keys under the analytical solution of the approximate dual problem. Thus, the authors demonstrate that the closed-form solution of the approximate dual problem is a generalized attention algorithm. Under the proposed framework, the design of preference distributions is flexible; for example, one can choose different types of context information depending on tasks.

In experiments, the authors discuss how much the deviation of the approximate dual's solution from the ground truth depends on \alpha.

**Limitations And Societal Impact:**

(Limitations)
In my understanding, Theorem 2 is incomplete, while it is supposed to be the major contribution of the paper. In Theorem 2, the approximation error depends on the remainders of the Taylor approximation of the cumulant generating function (CGF) of the preference distribution. In understanding, the characteristics of the CGF's higher-order Taylor terms are not trivial, except the normal distributions. Therefore, I consider it essential to specify the conditions of the remainders and how small \alpha should be to validate the approximation. One may only consider some common choices of preference distributions, such as the categorical or normal distributions. Note that the authors mention that in Lines 306-307, the approximations have implicitly assumed that the bounded moments of the preference distributions. However, I consider that Theorem 2 specify such conditions. Moreover, as far as I know, the behaviors of cumulants differ from moments in higher-order, so it is important to describe if preference distributions have bounded cumulants instead of bounded moments.

While the paper provides a novel interpretation of the attention algorithms, the discussion about testing the hypothesis (= the proposed framework) is limited. As I mentioned in the "Originality & Significance," one way to (indirectly) test if the proposed framework is generalized attention is to analyze if reducing the approximation errors improves the performance in a set of toy examples. I believe that providing such discussions will enormously improve the significance of the paper.



(Societal Impact)
N/A

**Main Review:**

Overall I enjoy reading the paper and believe the paper will be an essential contribution to the ML community. However, the paper should improve the quality and clarify.

(Originality & Significance)
I found that the proposed framework can provide a significant contribution to the ML community. For example, the authors can check if reducing the approximation errors improves the performance of the attention mechanisms. In the proposed framework, the authors take two levels of approximations. The first is to take the second-order Taylor polynomial of the cumulant generating function (CGF) in the dual problem. The second is the approximation from Eq 17 to Eq 18. If we focus on the latter, we notice that one can directly plug Eq. 17 into Eq. 14 without using Eq. 18. The solution contains the covariance of the keys, and the resulting attention naturally takes into account the correlations between keys. It would be exciting to see if the attention with the covariance terms will improve the performance.

Similarly, the authors can discuss the approximation errors of the second-order Taylor polynomial. It would be very interesting to see if the exact solution or higher-order polynomial will improve attention algorithms.


(Quality & Clarity)
The abstract and introduction section doesn't provide enough information about the paper and potential discussions. For example, it is difficult to guess what "a certain type of optimization" the authors refer to and in what context the authors introduce the generalization of the attention.

It is challenging to clarify how the proposed framework is supposed to look. I highly rely on the reference, G. Rioux et al., '20. I believe that the paper should be improved. For example, one may include 1) the motivation of solving the dual instead of the primal, or 2) what conditions described in the proposed method provide strong duality.

While Section 3-4 provides example cases in attention mechanisms, it isn't easy to draw a connection from the descriptions to the proposed framework.

In my understanding, the connection of the proposed framework to the principle of maximum entropy is invalid. There must be a historical reason for naming "maximum entropy on the mean (MEM)." Still, it would be nice to clarify the difference between the MEM and maximum entropy principle. I also found that G. Rioux et al., '20. also clarifies that MEM is different from the principle of maximum entropy (see Sec 3.1 in their paper)



(Minor comments)
- In Eq 17, a \alpha → 0, isn't \Sigma^{-1} z the only one left?

**Time Spent Reviewing:**

>12hrs

---

> ### Author Response · Authors · 2021-08-10
> **Clarification**
>
> Thank you very much for your insightful comments. Our responses follow.
>
> - The authors can check if reducing the approximation errors improves the performance of the attention mechanisms. In the proposed framework, the authors take two levels of approximations. The first is to take the second-order Taylor polynomial of the cumulant generating function (CGF) in the dual problem. The second is the approximation from Eq 17 to Eq 18. If we focus on the latter, we notice that one can directly plug Eq. 17 into Eq. 14 without using Eq. 18. The solution contains the covariance of the keys, and the resulting attention naturally takes into account the correlations between keys. It would be exciting to see if the attention with the covariance terms will improve the performance. Similarly, the authors can discuss the approximation errors of the second-order Taylor polynomial. It would be very interesting to see if the exact solution or higher-order polynomial will improve attention algorithms.
>
> **Response**. This is a nice suggestion. We have implemented an experiment performing an en-de translation task which indeed shows a better approximated $\lambda^*$ can improve the attention-based algorithm's performance. Please refer to our comment box entitled "Extra experimental results for reviewers" for details.
>
>
> - The abstract and introduction section doesn't provide enough information about the paper and potential discussions. For example, it is difficult to guess what "a certain type of optimization" the authors refer to and in what context the authors introduce the generalization of the attention.
>
> **Response**. In the revised version, we will provide more detailed information in the abstract and the introduction to ensure readers can grasp the main ideas of our paper as soon as possible.
>
> - It is challenging to clarify how the proposed framework is supposed to look. I highly rely on the reference, G. Rioux et al., '20. I believe that the paper should be improved. For example, one may include 1) the motivation of solving the dual instead of the primal, or 2) what conditions described in the proposed method provide strong duality.
>
> **Response**. We note that our work only uses G. Rioux et al.'s results stated in Theorem 1, namely, that the minimizer of Eq (6) takes the form stated in Eq (13). To understand the results of our paper, readers can treat Theorem 1 as a fact (i.e. black box). To offer a broader context to readers, however, we will briefly summarize the motivation and techniques behind G. Rioux's work, including MEM. Re (1) and (2): in a nutshell strong duality holds because the optimization problem is convex, and the solution has a nice characterization in the dual domain.
>
>
> - While Section 3-4 provides example cases in attention mechanisms, it isn't easy to draw a connection from the descriptions to the proposed framework.
>
> **Response**. The purpose of these examples is to give the reader some concrete settings - common in practice - where the assumptions of our optimization problem hold. We will try to polish the writing to make it more clear.
>
> - In my understanding, the connection of the proposed framework to the principle of maximum entropy is invalid. There must be a historical reason for naming "maximum entropy on the mean (MEM)." Still, it would be nice to clarify the difference between the MEM and maximum entropy principle. I also found that G. Rioux et al., '20. also clarifies that MEM is different from the principle of maximum entropy (see Sec 3.1 in their paper)
>
> **Response**. Thank you for pointing out this potential confusion. Indeed, the two concepts are different but closely related. We touched on this in L66 but will add a sentence to make this more clear. Briefly the Maximum Entropy principle postulates that the best guess of a distribution from which an i.i.d sample is drawn is the one that has the maximum entropy. MEM just regularizes this with a mean constraint, and as such is a special way of using the Maximum Entropy principle. Rioux et al also explain this in their Sec 3.1. While they take advantage of MEM for image deblurring, it is not yet widely known in machine learning and our hope is to introduce readers to its usefulness. Perhaps this resolves the confusion, or could you explain what you consider to be invalid? We were not clear on the concern you express. Thank you for flagging this issue though!
>
> - In Eq 17, as $\alpha \rightarrow 0$, isn't $\Sigma^{-1} z$ the only one left?
>
> **Response**. We are afraid not. Let us consider the 1D case, where $\Sigma^{(k)}$ is reduced to $\sigma^2$. Then Eq (17) becomes
>
> $$\lambda^* = \frac{\alpha}{1+\alpha \sigma^2} z^{(k)}.$$
>
> As
>
> $$\lim_{\alpha \rightarrow 0} \alpha z^{(k)} - \frac{\alpha}{1+\alpha \sigma^2} z^{(k)} =  \lim_{\alpha \rightarrow 0} \frac{\alpha^2\sigma^2}{1+\alpha \sigma^2} z^{(k)} = 0,$$
>
> we see that $\frac{\alpha}{1+\alpha \sigma^2} z^{(k)} \rightarrow \alpha z^{(k)}$ when $\alpha \rightarrow 0$.
>
>
>
> - In my understanding, Theorem 2 is incomplete, while it is supposed to be the major contribution of the paper. In Theorem 2, the approximation error depends on the remainders of the Taylor approximation of the cumulant generating function (CGF) of the preference distribution. In understanding, the characteristics of the CGF's higher-order Taylor terms are not trivial, except the normal distributions. Therefore, I consider it essential to specify the conditions of the remainders and how small $\alpha$ should be to validate the approximation. One may only consider some common choices of preference distributions, such as the categorical or normal distributions. Note that the authors mention that in Lines 306-307, the approximations have implicitly assumed that the bounded moments of the preference distributions. However, I consider that Theorem 2 specify such conditions. Moreover, as far as I know, the behaviours of cumulants differ from moments in higher-order, so it is important to describe if preference distributions have bounded cumulants instead of bounded moments.
>
> **Response**. We will more explicitly state the assumption before introducing Theorem 2. We understand your concern for the difference between the bounded cumulant and bounded moments and acknowledge that it would be easier to derive Theorem 2 if we assume bounded cumulants. However, we note that by assuming bounded absolute moment, one obtains that cumulants are also bounded. For example, [Dubkov and Matakhov (1975)](https://link.springer.com/content/pdf/10.1007%2FBF01043479.pdf) show that
>
> $$
> \left|\kappa_{n}\right| \leq n^{n} E\left[|X-E[X]|^{n}\right],
> $$
>
> where $\kappa_{n}$ is the $n^{\rm th}$ cumulant. We would continue assuming bounded moment as it can be easily implied from the bounded templates in $\mathbf{T}$.
>
> Obtaining a closed-form expression for approximation error (of the CGF) even in terms of error in $\lambda$ would be highly technical. To go further and express this explicitly in terms of $\alpha$ would require extra assumptions on the probability's distribution and its support, which we think would be a major digression for this paper and is more proper to be left for future work.

---

> > ### Comment · Reviewer_ZmjE · 2021-09-01
> > **Reply to the authors' response**
> >
> > I apologize for the late response. Thanks to the authors for the thorough responses.
> >
> > First, I raise a concern that the hypothesis testing of the proposed interpretation is necessary, and the author provides "Extra experimental results for reviewers". In the experiments, the results demonstrate that reducing the approximation error of the dual problem improves the results. I'm excited to hear that the experiment results support the hypothesis that the "(weighted) attention algorithm" can be interpreted as an approximate solution to a density estimation problem.
> >
> > Second, I point out that it is required to improve the description of the proposed method and relevant sections, including the introduction and abstract. The authors respond to improve writing in the revised version.
> >
> > Third, I conclude that Theorem 2 is incomplete. The author responds that (1) the cumulants of a random variable are bounded if the absolute moments are bounded, and (2) a closed-form expression for the approximation error will defer to future researches. In my understanding, mathematical theorems (or lemmas, propositions) should describe true statements. Thus it is important to include precise conditions, and thus one knows under which conditions the provided statement holds. In particular, it seems non-trivial to infer the bounds for the proposed problem, which emphasizes the importance of understanding the precise conditions. I agree that providing proper bounds can be left to future works; however, "theorizing" statements should be avoided in such a case.
> >
> > In conclusion, the authors respond to most of my concerns. Unfortunately, however, such modifications require major revision, and thus I will keep my current score. On the contrary, I strongly believe that the revised version will be a significant contribution to the ML community.
> >
> > (Minor comment) For clarification, I refer that the maximum entropy principle states distributions with the largest entropy under some constraints (or the optimization problem to find such distributions). For example, the normal distribution is the maximum entropy distribution when the mean and variance are specified. I understand that the maximum entropy on the mean (MEM) is named after its nature to minimize "relative entropy", which is KL divergence, under constraints. I wanted to point out that the "maximum entropy principle" is different from MEM (I can be wrong).

---

### Official Review · Reviewer_p7xB · 2021-07-18

**Rating:** 6
**Confidence:** 4

**Summary:**

This work constructs an optimization problem whose approximate solution results in a general form of attention. The optimization objective is derived from three assumptions: 1. the prior of the attention distribution has a support of the memory bank; 2. the likelihood of the query vector in the attention mechanism is roughly conforming to an isotropic Gaussian centered at the difference between the mean of the attention distribution and the mean of a prior distribution; 3. the standard deviation of the isotropic Gaussian is large. Then the mean of the posterior of the attention distribution is a general form of attention, with the dot attention scores reweighted by the prior probabilities. Under this framework, when the prior is a uniform distribution, dot attention is recovered; when the prior is defined by positions and word masks, the attention used in T5 is recovered.

**Limitations And Societal Impact:**

Yes

**Main Review:**

Strengths:
1. The math is elegant. Under the three simple assumptions, a general form of attention happens to be the mean of the posterior.
2. The general form of attention makes sense intuitively: T5 can be understood as one special case when position information is used, and there are also other works that can be viewed as special cases such as applying a content selector before attention (https://arxiv.org/pdf/1808.10792.pdf).

Weaknesses:
1. My main concern is the lack of intuition. While general attention follows from an optimization problem derived from the three assumptions, this formalism doesn't provide any more justification. In fact, I think directly proposing the general attention is more intuitive than going through all the math, especially considering that we are not treating the attention variable as a random variable or use its distribution in any sense (such as by marginalizing it, rather than simply using its mean under this work's formalism).
2. I don't think it's reasonable to assume the standard deviation of the isotropic Gaussian to be large ($\alpha$ being small under this paper's notation). There are situations where we use a high $\alpha$. For example, there are cases when we use a low "temperature" in attention to get more spiky or even one-hot distributions (such as in Gumbel-softmax https://arxiv.org/pdf/1611.01144.pdf). While this paper argues that the prior distribution reduces the support of the attention from tens of thousands of words to several dozens, I don't think "the inference of p(k) should strongly anchor around this prior information" (L237): no matter how large $\alpha$ is, the optimal solution of Eq. 6 always has the same support as the prior, so using whatever $\alpha$ is always anchoring around this prior information if we just want to have the same support.

Suggestions:
1. I think maybe it'd be more clear to get rid of the superscript $^{(k)}$ and simply mention that only a single step is considered in the case of predicting a sequence.
2. The motivating example is on a continuous case while the focus of the rest of the paper is on a discrete case. I think maybe you don't really need this motivating example.
3. The discussion section mentioned that the proposed approach might generalize to continuous cases. I think this paper would be stronger if you can make that work (and the motivating example would make more sense).

Overall, this paper is well-written and the math is elegant. However, the intuition and motivation of the objective function are not very clear and I am not convinced that having this objective function provides a principled justification for attention. Therefore, I am leaning towards rejecting this paper unless I missed anything.

==== Post Rebuttal ====

My concern about the assumption $\alpha$ being small has been resolved: it makes sense that in practice (transformers) $\alpha=\frac{1}{\sqrt{d}}$ is small.

Besides, it's interesting to see the extra experiment that shows optimizing 6) has a similar BLEU score as using $\alpha z^{(k)}$, and that $\alpha z^{(k)}$ is close to the optimal solution.

Therefore, I'm increasing my score to 6. That being said, I think this paper still can be further improved: for example, the BLEU score in the added experiment is quite low (I'd suggest using a smaller translation dataset such as IWSLT14 De-En), and I think it would be much more convincing if you can just use a trained transformer (with normal attention), and show that optimizing 6) gets a similar BLEU score. Lastly, I think this paper would be much stronger if you can add some experiments with a continuous version.

**Time Spent Reviewing:**

5

---

> ### Author Response · Authors · 2021-08-10
> **Clarification**
>
> Thank you very much for your insightful comments. Our responses follow.
>
> - *My main concern is the lack of intuition. While general attention follows from an optimization problem derived from the three assumptions, this formalism doesn't provide any more justification. In fact, I think directly proposing the general attention is more intuitive than going through all the math, especially considering that we are not treating the attention variable as a random variable or use its distribution in any sense (such as by marginalizing it, rather than simply using its mean under this work's formalism).*
>
> **Response**. We agree that the structure of general attention is sensible at an intuitive level. However, if it is derived purely heuristically, it is not evident how to interpret and adjust the architecture's design details. By contrast, our derivation shows that applying the softmax function for the template weights corresponds to using KL-divergence to measure the similarity of the preference distribution and the one being estimated. Besides, our derivation also shows that the selection of parameter $\alpha$ depends on how accurate the original estimation $\mu^{(k)}+z^{(k)}$ of $h^{(k)}$ is (see Eq (10)). Such insights can hardly be given if we design the block heuristically.
>
> More importantly, our interpretation of attention suggests the existence of more general and powerful attention structures. For example, as discussed in our paper, using KL-divergence in the convex optimization problem (see Eq (6)) requires the estimated distribution to have the same support as the preference distribution. This property may not be desired in many tasks. For instance, in translation tasks, the target sentence is unchanged if we replace some words in the source sentence with their synonyms (like replacing "good" with "nice"). Viewed through the lens of our optimization formulation, we notice this undesired property of attention as arising from the use of KL divergence in the regularization term, which in turn requires the model to put zero weights on words not appearing in the source sentence. Moreover, with our formulation, we see this problem can in fact be solved by replacing the KL divergence with an optimal transport-based measure that accounts for word similarities in their embedding space. This work is being prepared for submission to another conference.
>
> Regarding the concern of treating the normalized weights as a random variable's distribution, we acknowledge that this perspective is not very popular. But there is related work adopting this perspective (e.g. "hard attention" in [Show, Attend and Tell: Neural Image Caption Generation with Visual Attention](http://proceedings.mlr.press/v37/xuc15), Google Scholar Citations 7408). Essentially, we treat attention weights as a distribution just for the convenience of using some related expressions/definitions originally introduced in Statistics (like KL divergence, MGF, cumulant, etc.).
>
> - *I don't think it's reasonable to assume the standard deviation of the isotropic Gaussian to be large ($\alpha$ being small under this paper's notation). There are situations where we use a high $\alpha$. For example, there are cases when we use a low "temperature" in attention to get more spiky or even one-hot distributions (such as in Gumbel-softmax https://arxiv.org/pdf/1611.01144.pdf).*
>
> **Response**. This is a valid concern. We agree that our method cannot cover the case that $\alpha$ is large, and we will make this point clearer in the revised paper. We want to note that, in most of the practical models adopting attention, $\alpha = \frac{1}{\sqrt{d}}$ is small ($d$ is the dimension of the attention's heads and usually takes a large value). Thus, our analysis at least applies to a major portion of attention-based models.
>
> Regarding the Gumbel-softmax paper, we do not find its direct connection to our results as it focuses on solving backpropagation problems in stochastic neural networks. We wonder if you could kindly provide extra details on this part.
>
> - *While this paper argues that the prior distribution reduces the support of the attention from tens of thousands of words to several dozens, I don't think "the inference of p(k) should strongly anchor around this prior information" (L237): no matter how large $\alpha$ is, the optimal solution of Eq. 6 always has the same support as the prior, so using whatever $\alpha$ is always anchoring around this prior information if we just want to have the same support.*
>
> **Response**. We agree that regardless of the value of $\alpha$, the KL divergence requires the support of the optimal solution to be a subset of the prior's. We notice that our statement here might be a bit confusing. The message we try to deliver is that: the squared norm term contains significantly less reliable information than the KL divergence term and thus should be considered less throughout the estimation. Although decreasing $\alpha$ cannot give extra strength to the support selection, it can indirectly decrease the weight of the squared norm term in Eq (6) so that the term has less effect on the final estimated distribution. We will rewrite this sentence in the revised version to avoid any potential confusion.
>
>
> - I think maybe it'd be more clear to get rid of the superscript (k) and simply mention that only a single step is considered in the case of predicting a sequence.
>
>
> **Response**. We will remove the superscript in the revised version.
>
>
> - *The motivating example is on a continuous case while the focus of the rest of the paper is on a discrete case. I think maybe you don't really need this motivating example.*
>
> **Response**. We understand your concern. We start with a continuous toy example because it has much simpler expressions and contains many key ingredients that we will use in later more complex discussions. With this example we wish to offer readers an overview of our analysis workflow, to help build an understanding of our main results. Our analysis (up to Eq (19)) applies to the continuous case, but we have not found any practical scenario of continuous attention to support our theoretical results (Please also refer to our subsequent response on this issue). Due to this concern, we decided to mainly focus on the discrete case to keep our discussion succinct and easy to follow, and we leave the immediate generalization to the continuous version for future work that should become practically relevant.
>
> - *The discussion section mentioned that the proposed approach might generalize to continuous cases. I think this paper would be stronger if you can make that work (and the motivating example would make more sense).*
>
> **Response**.  You are right. Our theoretical analysis in fact applies to the continuous case. Almost all the attention architectures we have seen today, however, are discrete. Since the main focus of our paper is to understand the existing attention mechanism and show that it can be seen as a solver of a family of estimation problems, we focus on the discrete case and leave the continuous version for future work in the interests of cleaner discussions and clear messages delivered by this manuscript.

---

> > ### Comment · Reviewer_p7xB · 2021-09-01
> > **response taken**
> >
> > Thanks for the response!
> >
> > My concern about the assumption $\alpha$ being small has been resolved: it makes sense that in practice (transformers) $\alpha=\frac{1}{\sqrt{d}}$ is small.
> >
> > Besides, it's interesting to see the extra experiment that shows optimizing 6) has a similar BLEU score as using $\alpha z^{(k)}$, and that $\alpha z^{(k)}$ is close to the optimal solution.
> >
> > Re my point about Gumbel-Softmax: yes there's no direct connection, I just want to make a point that there are cases where we want spiky attentions by scaling the dot product results with a "temperature" parameter, which is equivalent to increasing $\alpha$ and contradicts the assumption that $\alpha$ is usually small. However, I can also understand that normal attention is still the most widely used attention formulation.
> >
> > Therefore, I'm increasing my score to 6. That being said, I think this paper still can be further improved: for example, the BLEU score in the added experiment is quite low (I'd suggest using a smaller translation dataset such as IWSLT14 De-En), and I think it would be much more convincing if you can just use a trained transformer (with normal attention), and show that optimizing 6) gets a similar BLEU score. Lastly, I think this paper would be much stronger if you can add some experiments with a continuous version.

---

### Author Response · Authors · 2021-08-10
**Extra experimental results for reviews**

To address the questions of some reviewers, we performed additional experiments. The objective of these experiments is to show
1. that an improved approximation of the solution $\lambda^*$ to optimization problem (6) improves an attention-based model's performance, and
2. that the approximation of $\lambda^*$ for regular attention deviates only slightly from the optimal $\lambda^*$.

We describe the experiments here and refer to them in our responses to individual reviewers.


According to Theorem 1, the exact $\lambda^*$ that should be plugged into Eq (13) is the maximizer of


$$ \max_{\lambda \in \mathbb{R}^{d}}\left\\{\left\langle\lambda, \mu^{(k)}+z^{(k)}\right\rangle-\frac{1}{2 \alpha}\\|\lambda\\|^{2}-\log \int_{\mathbb{R}^{d}} u^{(k)}(\mathbf{a}) \exp \langle\mathbf{a}, \lambda\rangle \mathrm{d} \mathbf{a}\right\\} \hspace{1em} ({\rm 1})$$


As the exact $\lambda^*$  does not have a closed-form expression, we have shown that it can be approximated by $\alpha z^{(k)}$ when $\alpha$ is small. Plugging the approximated $\lambda^*$ into Eq (13) yields the regular dot-product attention structure. In this experiment, we test if we can gain extra performance if $\lambda^*$ is more accurately approximated.

We use a gradient ascent algorithm to improve the approximation accuracy. Let
$$g(\lambda) = \left\langle\lambda, \mu^{(k)}+z^{(k)}\right\rangle-\frac{1}{2 \alpha}\\|\lambda\\|^{2}-\log \int_{\mathbb{R}^{d}} u^{(k)}(\mathbf{a}) \exp \langle\mathbf{a}, \lambda\rangle \mathrm{d} \mathbf{a}.$$
We set  $\lambda_0 = \alpha z^{(k)}$ and $\lambda_{k+1} = \lambda_k + \eta_k\nabla g(\lambda_k)$, where $\eta_k$'s are trainable step sizes and are different in various gradient ascent iterations. We repeat the iteration for three times to get an improved approximation $\lambda_3$ and plug it into Eq (13) to get $\hat{h}^{(k)}$.


To implement our experiments, we adopt the BASE model used in paper [Attention is All You Need](https://arxiv.org/abs/1706.03762) and use the T5 transformer's positional encoding instead of the sin-cos one. We refer to the model with $\lambda^*$ approximated by $\alpha z^{(k)}$ as *Attn* model and the one with $\lambda^*$ approximated by $\lambda_3$  as *Attn_rec* model.

We implement our experiment by performing the WMT14 en-de translation task with the same validation and test datasets. Due to the limited access to computation resources, we train the models using the first 250,000 samples in the training set (roughly 5.55% of the whole training data). We train both models for $10$ epochs and list their valid loss and valid BLEU for each epoch:


| Epoch | Attn (valid loss) | Attn_rec (valid loss) | Attn (valid BLEU4) | Attn_rec (valid BLEU4) |
|-------|-------------------|-----------------------|-------------------|-----------------------|
| 1     | 10.595            | 10.573                | 0.52              | 0.49                  |
| 2     | 9.48              | 9.446                 | 0.83              | 0.95                  |
| 3     | 8.646             | 8.596                 | 2.09              | 2.18                  |
| 4     | 8.646             | 7.917                 | 3.72              | 4.00                  |
| 5     | 7.432             | 7.388                 | 5.78              | 5.92                  |
| 6     | 6.997             | 6.908                 | 7.35              | 8.01                  |
| 7     | 6.685             | 6.561                 | 9.73              | 10.25                 |
| 8     | 6.488             | 6.424                 | 10.38             | 10.38                 |
| 9     | 6.475             | 6.278                 | 10.59             | 11.58                 |
| 10    | 6.354             | 6.216                 | 11.25             | 12.36                 |

We also test the models on the test dataset. Attn achieves BLEU4 9.11, and Attn_rec achieves BLEU4 9.43.

Our experimental results show that a more accurate approximation on $\lambda^*$ results in a faster validation loss drop, BLEU increase and a better test result, which to an extent supports our theoretical work.


Moreover, we have also checked if $\alpha  z^{(k)}$ in the trained Attn model gives a good approximation of $\lambda^*$. Our test is based on $250$ samples randomly picked from the Attn model's different attention layers when tested on the test dataset.  For each sample, we fetched the value $\alpha z^{(k)}$ that is served as the approximation of $\lambda^*$ in our theory. Also, we explicitly find $\lambda^*$ by implementing a gradient ascent algorithm on Eq(1). We then calculate $$\frac{||\lambda^* - \alpha  z^{(k)}||_2}{||\lambda^*||_2}$$
and report their distributions as follows:

    0.0   ....................
    0.1   ..........................................................
    0.2   ..................
    0.3   ....
    0.4
    0.5
As we can see, in most of the cases, the normalized approximation error is low, which implies $\alpha z^{(k)}$ is a good approximation of $\lambda^*$. This also indicates that our proposed optimization problem (6) provides a principle justifying the design of attention modules.

---

### Author Response · Authors · 2021-08-10
**General comments**

We wish to emphasize that our work provides a principled perspective by which to understand and design attention. It is, to our knowledge, the first contribution of this kind for attention. The overall philosophy of our work is to first isolate the attention module from a deep neural network and ask: what problem is this module solving? We show it solves a specific inference problem. Some examples of insight this produces: our derivation shows that applying the softmax function for the template weights corresponds to using KL-divergence to measure the similarity of the preference distribution and the one being estimated; our derivation also shows that the selection of parameter $\alpha$ depends on how accurate the original estimation $\mu^{(k)} + z^{(k)}$ of $h^{(k)}$ is (see Eq (10)); finally, our interpretation of attention suggests design strategies for more general and powerful attention structures.

---

> ### Comment · Area_Chair_sQtd · 2021-08-22
> **Please read author response**
>
> Dear reviewers,
>
> If you haven't done so yet, a friendly reminder to read the author response to your review and to update your score if necessary.
>
> Please also take a look at "Extra experimental results for reviews".
>
> Thanks,
> the area chair

---

### Decision · Program_Chairs · 2021-09-27

**Decision:**

Reject

**Comment:**

This paper received an overall score of 6 and is therefore borderline.

All reviewers agreed that it's a very promising paper that could shed some light on attention models.

However, in the current state, we believe the paper isn't ready for publication yet:

- The writing would benefit from some polishing.
- Section 3 "Motivating example" feels disconnected from the rest of the paper. We would personally remove it since it appears in Theorem 1.
- Section 4, Figure 1 and Figure 2 were difficult to follow. Some effort needs to be spent on clarity.
- Reading the paper, we are left with the feeling "so what?". Although a variational perspective of attention is interesting, we already know empirically that attention works. Coming up with benefits that only this new view brings is very important. To that end, the experiments that the authors posted in the rebuttal are promising.
- Reviewer ZmJe mentioned issues with Theorem 2

Overall, we agree with reviewer ZmJe

> In conclusion, the authors respond to most of my concerns. Unfortunately, however, such modifications require major revision, and thus I will keep my current score. On the contrary, I strongly believe that the revised version will be a significant contribution to the ML community.

We therefore believe that the ideas described in the paper are not yet clear enough to obtain the impact that they deserve. Polishing the paper and adding more experiments will greatly increase the impact of this paper and the authors will eventually benefit from it.